# Low Colon Capsule Endoscopy (CCE) False Negative Rate for Polyps Excluding Reader Error

**DOI:** 10.3390/diagnostics13010056

**Published:** 2022-12-25

**Authors:** Serhiy Semenov, Conor Costigan, Mohd Syafiq Ismail, Deirdre McNamara

**Affiliations:** 1Trinity Academic Gastroenterology Group, Trinity Centre, Tallaght Hospital, Trinity College Dublin, D02 R590 Dublin, Ireland; 2Department of Gastroenterology, Tallaght University Hospital, D24 NR0A Dublin, Ireland

**Keywords:** colon capsule endoscopy, false negative rates, colonic polyp, capsule endoscopy

## Abstract

Background: CCE is a diagnostic tool lacking clinical data on false negative rates. We aimed to assess this rate and the reader/technical error breakdown. Methods: False negative CCEs were identified after comparing to a colonoscopy database. Missed pathology characteristics and study indications/quality were collated. Cases were re-read by experts and newly identified lesions/pathologies were verified by an expert panel and categorised as reader/technical errors. Results: Of 532 CCEs, 203 had an adequately reported comparative colonoscopy, 45 (22.2%) had missed polyps, and 26/45 (57.8%) reached the colonic section with missed pathology. Of the cases, 22 (84.6%) had adequate bowel preparation. Indications included 13 (50%) polyp surveillance, 12 (46%) GI symptoms, 1 (4%) polyp screening. CCE missed 18 (69.2%) diminutive polyps and 8 (30.8%) polyps ≥ 6 mm, 18/26 (69.2%) of these were adenomas. Excluding incomplete CCE correlates, colonoscopy total and significant polyp yield were 97/184 (52.7%) and 50/97 (51.5%), respectively. CCE total polyp and significant polyp false negative rate was 26.8% (26/97) and 16% (8/50), respectively. Following re-reading, reader and technical error was 20/26 (76.9%) and 6/26 (23.1%). Total and significant missed polyp rates were 20.6% (20/97) and 14% (7/50) for reader error, 6.2% (6/97) and 2% (1/50) for technical error. Conclusions: False negative CCE rate is not insubstantial and should be factored into clinical decision making.

## 1. Introduction

Colon capsule endoscopy (CCE) has been shown to be effective and is a recommended alternative to colonoscopy in certain clinical situations [1,2]. Several meta-analyses and systematic reviews have confirmed CCE’s accuracy in detecting colonic polyps [3,4,5,6,7,8,9]. In addition to superior polyp detection compared to CT colonography, there is growing evidence to suggest CCE may have a higher yield for colonic polyps over traditional colonoscopy, with previously deemed false positive CCEs being reclassified on a second look unblinded colonoscopy. One prospective screening CCE and colonoscopy comparison study, which routinely performed a “second look” un-blinded colonoscopy in patients with an initial false positive CCE, reported an increase in the overall adenoma yield of 16% [10].

As with colonoscopy, a low false negative rate is just as important as a high polyp yield, particularly as intervals for intermediate and low risk screening and surveillance subjects are extended with the introduction of new polypectomy guidelines [11,12]. Several CCE and colonoscopy comparison papers have reported the CCE false negative rate for polyps ≥ 6 mm, the cut–off recommended for onward referral for colonoscopy in current CCE guidelines [1]. In trials where both CCE and optical colonoscopy were both routinely performed within a reasonable timeframe, reported false negative rates for ≥ 6 mm polyps vary from 2% to 15.5% [13,14,15,16,17,18,19] (Table 1). Although difficult, polyp matching protocols were followed to account for lesions being miss classified rather than overlooked, and these rates probably reflect actual clinical experience. The false negative rate varies depending on lesion type, being higher for flat versus protruding or pedunculated lesions and for those in the right versus left colon [20,21]. In addition to capsule completion and bowel preparation, segment transit speed and in turn, adaptive frame rate, have also been identified as an independent risk factor for missing colonic lesions on CCE [19,22].

There is abundant evidence from colonoscopy practice to show that polyps can be overlooked for a variety of individual and technical reasons. Despite employing techniques to optimise complete mucosal imaging including distal caps, caecal retroflexion and patient positioning, to mention a few, there remains a recognised accepted inherent procedural risk of missing polyps [23,24,25,26,27]. Unlike colonoscopy, intra procedural manoeuvres cannot be applied to enhance the yield of CCE. However, technological advances including an expanded field of view (172°/camera), adapted frame rate technology and artificial intelligence have resulted in superior performance, with some evidence to show more technically advanced capsules may even have a higher polyp detection than colonoscopy.

In addition to the technical limitations of CCE, reader error or video misinterpretation can be a cause of overlooked polyps. CCE videos are recorded and are available post procedure, enabling easy repeat reading. This allows us the possibility to revisit false negative studies to determine if a missed polyp was a result of reader error or truly overlooked by the capsule, in other words, a technical error. A better understanding of false negative CCE tests will be beneficial. Establishing an expected false negative CCE rate and understanding its aetiology will help develop focused and tailored interventions while also providing more data to support the adoption of CCE specific quality standards as well as improving patient information and consent.

Our aim was to calculate the false negative rate of CCE for all polyps and polyps ≥ 6 mm detected on routinely scheduled colonoscopy in our patient cohort. To determine the proportion of missed polyps resulting from (a) reader error and (b) technical error.

## 2. Materials and Methods

Following ethical approval as a service evaluation initiative by the “process improvement” department which is part of the “quality safety and risk management directorate” in our centre, all adult patients routinely scheduled for both a CCE and colonoscopy over a 7 year period were identified from our capsule and endoscopy databases at Tallaght University Hospital, Dublin. All included cases were performed as part of CCE versus colonoscopy comparison studies for a variety of indications including FIT positive screening, lower GI symptoms, IBD assessment and polyp surveillance in our centre. All studies were performed with Medtronic (Minneapolis, MN, USA) PillCam Colon 2 capsules and read by trained CCE readers. Bowel preparation varied and reflected the best available practice at the time with the majority of cases receiving MoviPrep (Norgine, Mid Glamorgan, UK) based split-dose bowel preparation and booster regimen. Studies performed with first generation colon capsules, those where the interval between CCE and colonoscopy was more than 2 years, CCE’s performed for colonoscopy completion on the same day and incomplete colonoscopy studies were excluded. Incomplete CCEs were included in cases were the capsule reached a lesion later identified on colonoscopy. All colonoscopies were performed in a dedicated endoscopy unit in a tertiary teaching hospital.

Missed lesions, defined as a polyp or cancer of any size identified on colonoscopy without a corresponding lesion, matched for bowel segment and size, on CCE were identified. The overall false negative CCE rate, defined as the percentage of CCE studies with any overlooked polyp compared to the actual number of colonoscopies with confirmed polyps ($\frac{False\;Negatives}{False\;negatives+True\;Positives}$) was calculated. This was also calculated for significant polyps, defined as ≥6 mm. For false negative studies, patient demographics (age, gender, indication), procedural data (bowel preparation, etc.), lesion characteristics (location, size) and histology were documented. True positive studies were defined as CCEs with positive polyp correlation on colonoscopy. In comparing cohorts of false negatives to true positives, a “per study” approach was applied, where one study was a true positive if one or multiple significant polyps were later correlated on colonoscopy. In cases where multiple lesions were missed in the event of a false negative study or multiple lesions identified and correlated in true positive studies, location of the most significant/largest lesion was recorded. False positive CCE studies were also identified and compared to the remainder of positive CCEs who later had a positive correlate on colonoscopy. Further diagnostic test analysis was undertaken including CCE accuracy, sensitivity, specificity, positive and negative predictive metrics with the use of MedCalc Software Ltd, Version 20.118. diagnostic test evaluation calculator.

Subsequently, false negative CCE studies were anonymised and reread by an expert CCE reader, defined as >200 CCE cases performed, excluding training. Pillcam Rapid Reader Version 9 was employed in this study. The readers were unblinded to either the colonoscopy or initial CCE result. An expert panel, made up of 1 consultant gastroenterologist and 2 non consultant hospital doctors who were independent experienced capsule endoscopy readers, assessed any lesion not identified on the original report, or bookmarked on the original study file, which was found on the second reading. Only lesions approved by consensus by the panel were considered an initial reading error. Similarly, the panel also assessed negative reread studies, where the missed lesion was not identified on rereading. Studies where there were no corresponding lesion images were deemed technical errors, or true false negative studies, as opposed to reader error. Further chi square calculations on polyp characteristics associated with false negative events were undertaken.

## 3. Results

In order to establish true false negative cases, we assessed our CCE database and available colonoscopy reports. Of 532 CCE studies available on our database, 210 (39.5%) had a recorded follow up colonoscopy either scheduled after their CCE or booked for a separate reason within a 2 year period. Of the total 210 colonoscopies, 2 were excluded as they were incomplete studies and 5 had incomplete records, leaving 203 matched studies. Of all matched 203 colonoscopy reports, 45 (22.2%) studies registered polyps not seen on prior CCE report. Of these 45, 19 (42.2%) were excluded as false negative cases for incomplete CCE studies that did not reach the area where the polyp was seen on colonoscopy, i.e., controlling for true false negative events. The total matched CCEs with missed polyps resulted in 26. The mean interval between false negative cases and subsequent colonoscopies was 4.7 months (range 0–18 months). As previously mentioned, all included cases were performed as part of CCE versus colonoscopy comparison studies for a variety of indications including FIT positive screening, lower GI symptoms, IBD assessment and polyp surveillance in our centre.

To adequately calculate false negative rates, an assessment of true positive CCEs was undertaken. Overall, 97/184 (52.7%) patients had polyps identified on colonoscopy, after removing the 19 matched studies with incomplete CCEs. Of all CCE studies, 96/184 (52.2%) polyps were identified. Of the 184 CCEs, 25 (13.6%) were considered false positive following a full colonoscopy. This resulted in 71 (38.6%) true positive CCEs in our cohort. Out of 25 false positive cases, 6/25 (24%) of these were for CCE significant polyps that were not identified on subsequent colonoscopy. Repeat colonoscopies were not routinely performed on false positive CCE studies and further analysis on this section of the population is beyond the scope of this study.

There were 26 cases identified with an overlooked polyp on CCE, giving a false negative CCE rate for any polyp in our cohort of 26.8% (26/97). Examples of polyps seen on CCE re-reading are shown in Figure 1. Of these 18 (69.2%) were adenomas on histological assessment. None of the cases had high grade dysplasia on histology. No cancers were identified in this study. The mean age of cases was 62 years (range 26–81 years) with 54% (14/26) men and 46% women (12/26). The indication for CCE and colonoscopy was polyp surveillance in 13 (50%), lower GI symptoms in 12 (46%) and FIT screening in 1 (4%). An equal proportion of missed polyps were located in the right and left [13] colon where left sided lesions were seen distal to the splenic flexure. In the majority of cases (85%, 22/26) the quality of bowel preparation was deemed good or excellent and was not a contributory factor following assessment by the expert panel. False negative study demographics are seen in Table 2. After including incomplete CCEs, the overall accuracy of CCE in our cohort was 65.52% (sensitivity 61.21%, specificity 71.26%, positive predictive value 73.96%, negative predictive value 57.94%) as seen in Table 3.

Of the 97 patients with any polyp on colonoscopy, 50 (51.5%) had a significant polyp based on size (≥6 mm). Of missed polyps on CCE, 8/26 (30.8%) were considered significant giving a false negative CCE rate for significant polyps of 16% (8/50). There were no cancers overlooked in this cohort.

Comparing false negative with true positive studies including patients with polyps seen in both the right and left colon, revealed a tendency of CCE to miss right sided polyps more than left sided polyps, 43.3% (13/30) vs. 19.4% (13/67), *p* = 0.0070. False negative CCEs were more likely due to smaller polyps versus larger ones following a comparison with true positive colonoscopies, 38.3% (18/47) vs. 16% (8/50), *p* = 0.0066 (Table 4). No data on bowel preparation, transit times or polyp morphology for true positive cases was available for analysis.

After re-reading by a trained capsule endoscopist and expert panel review of false negative studies, reader error was the cause of the majority of missed lesions on CCE. In 20 /26 (77%) cases, images of missed polyps were identified on repeat video analysis and confirmed by the expert panel. While in only 6/26 (23%) cases no images were found corresponding to the missed lesion seen on colonoscopy, all bar 1 of which were diminutive <6mm. In total, missed lesions were a result of a technical error in 6.2% (6/97) and reader error is 20.6% (20/97) of our cohort. With respect to significant lesions, 2% (1/50) were due to technical error and 14% (7/50) due to reader error (Figure 2).

## 4. Discussion

In our study, the overall false negative CCE rate for any polyp was 26.8% and the false negative rate for significant polyps (≥6 mm) was 16%, which is in keeping with previous reports, suggesting the quality of CCE performance overall was acceptable and in line with published data from other centres. Similar to previous studies, proximal colon polyps are more likely to be missed. Although the design of our study, which excluded incomplete CCE studies which did not reach a missed lesion usually in the left colon could account for this disparity. Similarly, incomplete CCEs have been shown to be associated with poor quality bowel preparation and slow transit, and as such further subgroup analysis of these factors in our cohort was not deemed appropriate.

Although there is some evidence available to suggest reading errors are responsible for some false negative CCE it is not yet clearly established what proportion are a result of reader error rather than a true technical failure of the capsule to capture any image of the missed polyp. Spada et al. reported that of 7 false negative CCEs in 45 patients with polyps, 3 were simply misclassified based on size, and of 4 significant (>6 mm) missed lesions 50% (*n* = 2) were identified on repeat unblinded reading of the videos [18]. Our data suggest that reader error could be responsible for an even higher proportion of missed polyps, 77% (20/26) and that, particularly for significant polyps ≥6mm in size, that would trigger onward referral for colonoscopy and polypectomy, true capsule error is uncommon, 2% (1/50). This compares very favourably to colonoscopy with reported missed rates for any polyp and for advanced polyps in a recent systematic review and meta-analysis of 26% and 9%, respectively [28].

We report the miss rate for all polyps irrespective of size. This is relevant clinically as both size ≥ 6 mm and number > 3 irrespective of size are indicators for onward referral for colonoscopy. Overall, the number of misclassified CCE lesions based on size alone in our cohort was small (8/26, 30.8%), of significant lesions only 1 was a technical capsule error and 7 (87.5%) were missed on reading. We found that right sided polyps were more likely to be overlooked which correlates with previous studies as mentioned above. Unsurprisingly, a false negative study was more likely due to smaller polyps when compared to true positive cases. Overall, the small numbers involved make further interpretation of these findings difficult. Suffice to say, reading errors including both sizing errors and/or simply overlooking lesions warrant the development of specific targeted interventions to reduce their occurrence.

Although a standard reading protocol was not followed for rereading nor all videos with missed lesions reviewed by the same person, all positive lesions were verified by a panel of experienced readers which controlled for a potential experience disparity amongst the readers. Importantly all negative videos, which failed to identify the polyp seen on colonoscopy, were reviewed by all panel members as well, making it unlikely that the polyp was overlooked and that the true capsule error is reported.

We acknowledge that our study has inherent bias given its retrospective design and the overall number of missed lesions was small, however, our study highlights the importance of promoting techniques to enhance reading performance, such as further refining and defining optimal reading format and speeds, similar to performance recommendations for small bowel capsule endoscopy. It is encouraging re-reading of rapid transit segments and adoption of image enhancement reading modes including artificial intelligence (AI), among others to optimise polyp detection. While capsule AI technology is developing and is likely to improve both colonoscopy and CCE quality and polyp detection, with one systematic review suggesting sensitivities for polyp detection as high as 81.3–98.1% [29], for now it is not mainstream [30]. We also acknowledge that including colonoscopies up to 2 years post CCE may overestimate the false negative rates of CCE and we hope to repeat our assessment with ever increasing volume of CCE procedures being performed in our centre or possibly combining data from other capsule endoscopy centres, in order to mitigate this potential bias. It is worth noting that we excluded incomplete CCEs from the total technical error group which in turn reduced the false negative rate, however, the purpose of our study was to assess the potential polyps that would have been missed outright without the safety net of a routine completion sigmoidoscopy or colonoscopy (booked following an incomplete CCE).

The importance of training and capsule endoscopy reader experience has been highlighted in the literature [31] with one study reporting that negative predictive values rose with cases reviewed but did not continue increasing after the first 100 reads in a small bowel capsule endoscopy cohort [32]. Even though, in our case, calculating the level of reader experience of the original CCEs reports was outside the scope and power of our study, it is worth noting that in our centre, all studies are signed off at a weekly consultant led meeting which may, to a degree, negate the lower negative predictive rate of less experienced readers. In summary, when CCE is routinely available, it will not be fool proof and the ultimate responsibility will remain with the human operator. As such the relevance and need for good reading practice will remain and is highlighted in our study.

Despite the inherently difficult polyp matching process and accounting for multiple patient, procedure and reader factors, our study sheds light onto CCE as a valid alternative to colonoscopy in lower GI evaluation. CCE appears to have the benefit of comparable polyp detection rates and false negative rates without the associated risks of an invasive procedure. Knowing the likelihood of a missed polyp and its implications can help physicians make more informed decisions for their patients as well as increasing patient information at time of consent.

## 5. Conclusions

CCE is a viable alternative to colonoscopy for certain clinical indications, such as polyp surveillance, assessing GI reported symptoms, etc. As with colonoscopy, there is an inevitable false negative CCE rate, which should be factored into any clinical decision process. Further studies looking into CCE performance and rates of missed lesions could further aid clinical decision making, in particular, with the increased prevalence of AI capable capsules.

## Figures and Tables

**Figure 1 diagnostics-13-00056-f001:**
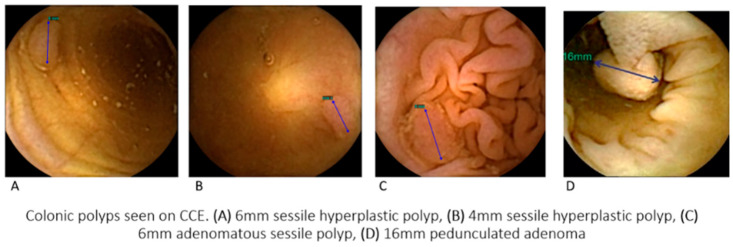
Polyps seen on CCE re-reading.

**Figure 2 diagnostics-13-00056-f002:**
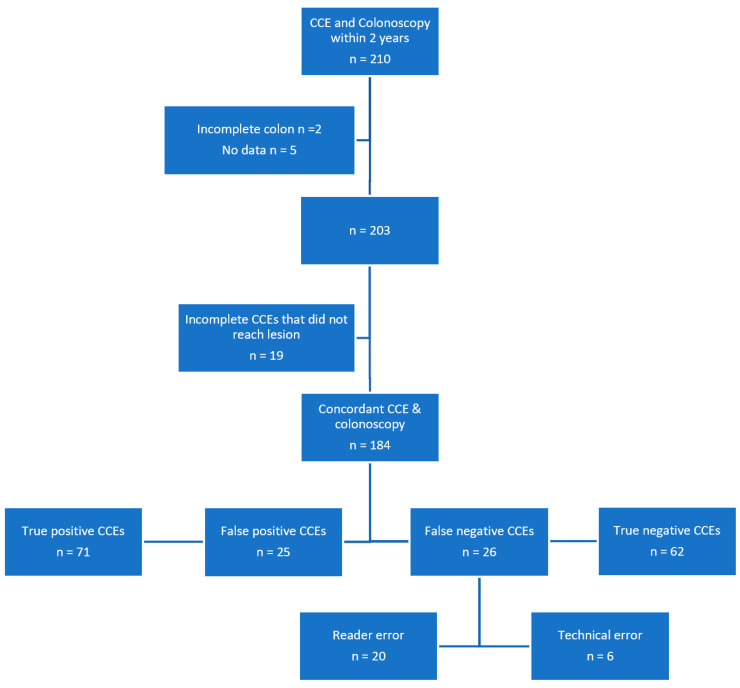
Study population.

**Table 1 diagnostics-13-00056-t001:** CCE False Negative Rate for polyps > 6 mm compared to optical colonoscopy.

Paper	False Negative Rate %	N
Pecere S. et al. [13]	10	6/60
Rex D.K. et al. [14]	13	25/192
Eliakim R. et al. [15]	11	2/16
Ismail M.S. et al. [16]	0	0/7
Holleran G. et al. [17]	12.5	2/16
Spada C. et al. [18]	15.5	7/45
González-Suárez B. [19]	1.3	1/78

**Table 2 diagnostics-13-00056-t002:** False Negative CCE Characteristics.

	Number	Percentage
**Overall**	26	%
** *Indication* **		
Polyp Surveillance	13	50%
Symptom investigation	12	46%
FIT (+) Screening	1	4%
** *Histology* **		
Hyperplastic	8	31%
Adenoma	18	69%
** *Location* **		
Proximal	13	50%
Distal	13	50%
** *Size* **		
<6 mm	18	69%
≥6 mm	8	31%
** *Bowel preparation* **		
Adequate (good or excellent)	22	85%
Inadequate (fair and poor)	4	15%

**Table 3 diagnostics-13-00056-t003:** CCE evaluation as a diagnostic test.

Statistic	Value	95% CI
Sensitivity	61.21%	51.72% to 70.11%
Specificity	71.26%	60.57% to 80.46%
Positive Likelihood Ratio	2.13	1.48 to 3.06
Negative Likelihood Ratio	0.54	0.42 to 0.71
Disease prevalence	57.14%	50.03% to 64.05%
Positive Predictive Value	73.96%	66.43% to 80.30%
Negative Predictive Value	57.94%	51.39% to 64.23%
Accuracy	65.52%	58.54% to 72.03%

**Table 4 diagnostics-13-00056-t004:** Polyp characteristics associated with false negative events.

	False Negative (FN) Cases	True Positive Cases	Characteristic Associated with FN
Total cases	26 (26.8%)	71 (73.2%)	
**Location:**			
Right sided polyps	13 (43.3%)	17 (56.7%)	Right sided ***p* = 0.0070**
Left sided polyps	13 (19.4%)	54 (80.6%)	
**Size:**			
<6 mm	18 (38.3%)	29 (61.7%)	<6 mm ***p* = 0.0066**
≥6 mm	8 (16%)	42 (84%)	

## Data Availability

Not applicable.

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
