# Peer review of "Low Colon Capsule Endoscopy (CCE) False Negative Rate for Polyps Excluding Reader Error"

_diagnostics, 2022, doi:10.3390/diagnostics13010056_

Round 1
Reviewer 1 Report
I consider adding some more points to the article:
- I found it difficult to follow the results presented in the current way. I suggest making it more clear and adding more charts,
- Correlation between capsule transit time and polyp detection,
- Correlation between missed polyps at capsule endoscopy and location of the missed polyps, and also between the size of missed polyps and their location,
Regarding the conclusions, I am not sure that the point of 77% of missed cases being the result of reading error is a good aspect. It can also reflect the lack of experience of the initial readers.
Reviewer 2 Report
The colon capsule endoscopy is a relatively new modality for colon screening. In the present study, the researchers investigated the false negative rate for polyps of colon capsule endoscopy.
The researchers found a significant rate of false negatives rate, which was 26.8%. After re-reading by a trained capsule endoscopist and expert panel review of false negative studies, reader error was the cause of the majority of missed lesions on CCE. Missed lesions were small and in the right colon.
The are several significant limitations of the study.
1. The abstract: includes many numbers and percentages with difficulties in following these numbers.
2. Introduction: The authors presented the data published in the literature in general and literature related to the false negative rates of colon capsule endoscopy. In addition, technical challenges were presented.
3. Important information not reported in the Materials and methods section, such as study population, Data collected, and Statistics analysis.
4. The expert panel is made up of 1 consultant gastroenterologist and 2 gastroenterology trainees, I am wondering why two trainees are included in the expert panel.
5. Results section: in table 3, the percentage should be given in addition to the numbers
6. Discussion section, no mention of the significant study limitations.
In summary, the present study focused on the false negative rate of the colon capsule endoscopy, which is a very important issue, but this specific study is of significant limitations; in addition to the above-mentioned points, the very small number of patients (45 studies with polyps not seen in the capsule, and of these 19 were excluded for non-reaching the area where the polyp was seen on colonoscopy), and of these 45 makes it very difficult to reach any conclusion that can be helpful and useful for the clinical practice.
Reviewer 3 Report
Thanks for the detailed paper, I have read it. I suggest that the author should revise the manuscript before resubmission. My main concern and comments are listed as follows.
1. The introduction section needs to be rewritten with much better motivation and providing the context for this work. It should include:
(1). Contextualization
(2). Importance/Relevance of the theme
(3). Research question
(4). Objectives
(5). Structure of the article
2. In order to increase the readability of the article, please add the main contribution of this article in the Introduction section.
3. The Literature Review section is missing. The author should present a critical review of various state-of-the-art methods.
4. It is better to use more evaluation metrics such as Accuracy, Precision, Sensitivity, Specificity, and F1-score.
5. Please add a table and compare the proposed work with deep learning algorithms and other methods in the Discussion section.
6. Please add the future scope of the current work in the Conclusions section?
7.Hyperspectral or multispectral techniques can also be used for the analysis of biological samples (such as the following references). Please discuss these aspects.
[1]"In‐vivo multispectral video endoscopy towards in‐vivo hyperspectral video endoscopy." Journal of biophotonics 10.4 (2017): 553-564.
[2] "A clinically translatable hyperspectral endoscopy (HySE) system for imaging the gastrointestinal tract." Nature communications 10.1 (2019): 1-13.
[3] "Open-source mobile multispectral imaging system and its applications in biological sample sensing." Spectrochimica Acta Part A: Molecular and Biomolecular Spectroscopy 280 (2022): 121504.
[4]Smartphone imaging spectrometer for egg/meat freshness monitoring. Analytical Methods, 14(5), 508-517.
Round 2
Reviewer 1 Report
The paper is now improved.
Author Response
Many thanks on behalf of all the authors,
Serhiy Semenov
Reviewer 2 Report
All points raised by me have been answered
Author Response

(The authors gave the same response as above.)
